# Model Merging
# in Pre-training of Large Language Models

**Yunshui Li, Yiyuan Ma, Shen Yan, Chaoyi Zhang, Jing Liu, Jianqiao Lu, Ziwen Xu**
**Mengzhao Chen, Minrui Wang, Shiyi Zhan, Jin Ma, Xunhao Lai, Yao Luo, Xingyan Bin**
**Hongbin Ren, Mingji Han, Wenhao Hao, Bairen Yi, Lingjun Liu, Bole Ma, Xiaoying Jia**
**Xun Zhou, Liang Xiang, Yonghui Wu**
ByteDance Seed
`liyunshui@bytedance.com`

## Abstract

Model merging has emerged as a promising technique for enhancing large language models, though its application in large-scale pre-training remains relatively unexplored. In this paper, we present a comprehensive investigation of model merging techniques during the pre-training process. Through extensive experiments with both dense and Mixture-of-Experts (MoE) architectures ranging from millions to over 100 billion parameters, we demonstrate that merging checkpoints trained with constant learning rates not only achieves significant performance improvements but also enables accurate prediction of annealing behavior. These improvements lead to both more efficient model development and significantly lower training costs. Our detailed ablation studies on merging strategies and hyperparameters provide new insights into the underlying mechanisms while uncovering novel applications. Through comprehensive experimental analysis, we offer the open-source community practical pre-training guidelines for effective model merging.

## 1 Introduction

Modern large language models (LLMs) [Seed et al., 2025, Achiam et al., 2023, Guo et al., 2025, Team et al., 2024, Yang et al., 2024a] have demonstrated remarkable capabilities with widespread applications across diverse tasks. Despite their exceptional performance in fundamental tasks, LLMs still face several critical challenges, including the extensive pre-training costs, discounted effectiveness of domain-specific post-training, imprecisely-predictable performance scaling, as well as the instability of large-scale training. Model merging [Yang et al., 2024b], as a relatively young topic, presents a promising approach to alleviate these practical challenges.

Recently, the benefits of model merging have been primarily studied in the post-training stage, where several models fine-tuned on different downstream tasks are combined into a single but more versatile model [Ilharco et al., 2023, Zhou et al., 2024, Yu et al., 2024]. For example, using the DARE [Yu et al., 2024] method to merge WizardLM [Xu et al., 2024] with WizardMath [Luo et al., 2025] shows a significant performance enhancement on GSM8K [Cobbe et al., 2021], raising its score from 2.2 to 66.3. In contrast, research on model merging during the pre-training phase remains scarce. Such pre-training merging typically involves combining checkpoints from a single training trajectory, as explored in LAWA [Kaddour, 2022] which utilizes model merging to accelerate the LLM training. However, as the model and data scales dramatically, independent researchers struggle to evaluate model merging's impact on large-scale models, mainly due to limited access to intermediate checkpoints from extensive pre-training. Although DeepSeek [Liu et al., 2024a] and LLaMA-3 [Grattafiori et al., 2024] have both indicated their employment of model merging techniques for model development, detailed information regarding these techniques has not been publicly disclosed.

39th Conference on Neural Information Processing Systems (NeurIPS 2025).

In this work, we mainly focus on model merging during the pre-training stage, introducing Pre-trained Model Average (PMA), a novel strategy for model-level weight merging during pre-training. To comprehensively evaluate PMA, we trained a diverse set of LLMs of varying sizes and architectures from scratch, including Dense models Grattafiori et al. [2024] with parameters spanning from 411M to 70B, as well as Mixture-of-Experts (MoE) architectures Shazeer et al. [2017] with activated/total parameters ranging from 0.7B/7B to 20B/200B. We first investigate the performance impact of PMA and establish systematic evaluations across different phases of the warmup-stable-decay (WSD) learning schedule, which lately becomes a popular choice of $lr$ scheduler for LLM pre-training since Hu et al. [2024]. Experimental results demonstrate that model merging during the stable training phase yields consistent performance gains at different training steps. More remarkably, applying PMA at early-stage of the $cosine$-decay phase usually achieve comparable or even superior performance to their final-stage annealing counterparts. These findings suggest that during the extensively lengthy pre-training stage with constant $lr$, PMA can serve as a fast, reliable yet low-cost simulator for the annealed performance, enabling both faster validation cycles and significant computational savings.

Building upon our PMA framework, we first evaluate its performance with various prevalent merging strategies, including Simple Moving Average (SMA) Johnston et al. [1999], Weighted Moving Average (WMA) Perry [2010] and Exponential Moving Average (EMA) Hunter [1986]. Notably, our experiments demonstrate that the performance differences among these methods gradually become negligible. We further investigate how these important factors of PMA, namely, the interval between each merging checkpoint, the number of models involved in merging, and the size of the model, would affect merging performance. Our analysis reveals two important findings: First, the optimal merging interval exhibits a clear scaling relationship with model size. Second, incorporating more checkpoints in the merging process consistently improves performance once training is completed.

Furthermore, we also investigated whether PMA could produce more effective initialization weights for the consecutive continued training (CT) or supervised fine-tuning (SFT) Wei et al. [2022] stages to enhance the downstream model performance. We practically observed that entering CT and SFT stages with PMA applied could yield smoother *GradNorm* curves, which thus helps stabilize the training dynamics yet without harming the performance, compared to initializing these stages with the latest available checkpoint as usual. This finding inspire a novel application of model merging for training stabilization, which we dubbed as PMA-init. We demonstrate that in scenarios when the LLM training experiences severe irrecoverable loss spikes with broken training dynamics, applying PMA-init over $N$ preceding checkpoints to resume training, enables reliable recovery from unstable training trajectories.

In summary, our paper makes the following key contributions:

- We present the Pre-trained Model Averaging (PMA) strategy, a novel framework for model merging during LLM pre-training. Through extensive experiments across model scales (from millions to over 100B parameters), we demonstrate that merging checkpoints from the stable training phase produces consistent and significant performance improvements.

- We delved into novel applications of model merging for weight initialization (PMA-init), to help stabilize training process without harming the downstream performance, especially when it suffers from irrecoverable loss spikes with broken training dynamics. Through extensive experiments, we demonstrate the effectiveness of PMA-init on both CT and SFT stages.

- We also comprehensively ablated various model merging techniques with their associated hyper-parameters. Our findings offer the research community practical pre-training guidelines with effective model merging. Nevertheless, the low cost and rapid deployment of PMA also make it a reliable and economic monitor for the pre-training process, to flexibly simulate the ultimate model performance after annealing.

## 2   Related Work

Model merging is an emerging field undergoing rapid development, with diverse applications across various domains. Typically, model merging is implemented during the **post-training** phase [Ilharco et al., 2023, Zhou et al., 2024, Yu et al., 2024, Yadav et al., 2024], where multiple models fine-tuned

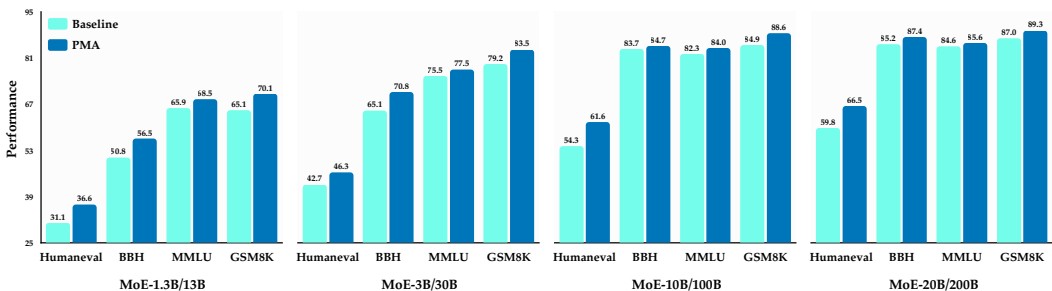

Figure 1: Comparison of downstream task performance for MoE models of varying sizes under stable training, before and after model merging.

on different downstream tasks are combined by merging their weights. This process effectively integrates the distinct capabilities of each individual model, resulting in a unified model that exhibits robust and comprehensive performance.

Recently, several methods have advanced this field significantly. For instance, Task Arithmetic [Ilharco et al., 2023], Ties-Merging [Yadav et al., 2023], and AdaMerging [Yang et al., 2024c] integrate Vision Transformer (ViT) models [Dosovitskiy et al., 2021] trained on distinct visual classification tasks, producing a single model capable of multi-task object classification. PAPA [Jolicoeur-Martineau et al., 2023] integrates the broad applicability of ensembling with the computational efficiency of weight averaging. MetaGPT [Zhou et al., 2024] frames model merging as a multi-task learning problem, aiming to minimize the average loss between the merged model and individual task-specific models. Fisher Merging [Matena and Raffel, 2022] employs a weighted fusion of model parameters, with weights determined by the Fisher information matrix. RegMean [Jin et al., 2023] elegantly addresses the merging process by formulating it as a linear regression problem solvable through closed-form solutions. Evolutionary-model-merge [Akiba et al., 2025] efficiently optimizes merging coefficients using evolutionary algorithms. Additionally, DARE [Yu et al., 2024] merges multiple task-specific language models into a versatile unified model by randomly dropping and subsequently rescaling the delta parameters.

However, research on model merging during the **pre-training** phase remains relatively limited. Such studies typically refer to incorporating checkpoints within a single training trajectory during large language model (LLM) pre-training. For example, LAWA [Kaddour, 2022, Hägele et al., 2024, Li et al., 2022] demonstrated that merging checkpoints at intermediate stages can significantly accelerate training. Sanyal et al. [Sanyal et al., 2024] further indicated that checkpoint averaging combined with a high learning rate in pre-training trajectories contributes to faster convergence. Additionally, Checkpoint Merging [Liu et al., 2024b] provided a comprehensive evaluation of the effectiveness of merging checkpoints at different stages during the pre-training of the Baichuan2 [Yang et al., 2023] LLM. Furthermore, technical reports of large-scale models such as Deepseek V3 [Liu et al., 2024a] and LLaMA3.1 [Grattafiori et al., 2024] also mention the use of model merging techniques during pre-training, although detailed methodologies have not been publicly disclosed. This paper primarily explores techniques for model merging within the pre-training paradigm. To the best of our knowledge, this is the first study to provide detailed technical insights into scaling model merging methods to significantly larger model sizes. We also discuss practical approaches for effective model merging and analyze its potential capabilities as well as its limitations.

## 3 Preliminaries

In this section, we describe the fundamental experimental framework, introduce the notations and concepts used in model merging, and present multiple variants of model merging techniques.

**Experimental setup.** In terms of model architecture, we independently trained a series of MoE and dense models. We employ a Warmup-Stable-Decay (WSD) learning rate scheduler Hu et al. [2024], which begins with a short warmup period, followed by an extended period of stable training at a constant learning rate, and concludes with annealing to a relatively small learning rate. The learning rates are determined according to scaling law guidelines Bi et al. [2024], Kaplan et al. [2020],

employing optimal values for training on an internal pretraining corpus comprising trillions of tokens. Although specific model architectures and datasets have not yet been publicly released, we posit that our findings are not strongly tied to these particular choices, as subsequent experiments primarily focus on MoE structures. Related conclusions for dense models are provided in the Appendix A. For evaluation, we primarily report results on open-source benchmarks in both few-shot and zero-shot settings, including: ARC-Challenge Clark et al. [2018], BBH Suzgun et al. [2023], DROP Dua et al. [2019], WinoGrande Sakaguchi et al. [2021], HellaSwag Zellers et al. [2019], MMLU Hendrycks et al. [2021], C-Eval Huang et al. [2023], TriviaQA Joshi et al. [2017], Ape210K Zhao et al. [2020], GSM8K Cobbe et al. [2021], MATH Zhao et al. [2020], MBPP Austin et al. [2021], HumanEval Chen et al. [2021], AGIEval Zhong et al. [2024], GPQA Rein et al. [2024], and MMLU-Pro Wang et al. [2024]. The weighted average score across these benchmarks serves as the model's comprehensive performance metric. Unless otherwise specified, we report this score as the model's performance metric to ensure evaluation reliability.

**Notions and concepts.** Our main focus is on model merging during pre-training, where the merged entities are sequential checkpoints along the training trajectory. Suppose we aim to merge $N$ models, with each model's parameters denoted as $M_i$ (where $i$ ranges from 1 to $N$). Each model has an associated weighting coefficient $w_i$, and the merged model $M_{avg}$ is computed as:

$$M_{\text{avg}} = \sum_{i=1}^{N} w_i M_i. \tag{1}$$

We assume that the data consumption of these models form an arithmetic sequence with a common difference $V$, formulated as:

$$V = T_{i+1} - T_i, \tag{2}$$

where $T_i$ represents the cumulative number of tokens consumed by the $i$-th model.

**Model merging variants.** Model merging techniques vary primarily in how they assign weights ($w_i$) to individual models. This paper examines three popular approaches for weight assignment, namely the Simple Moving Average (SMA), Exponential Moving Average (EMA), and Weighted Moving Average (WMA).

The first approach, *Simple Moving Average (SMA)*, treats all models equally. For instance, when combining 10 models, each model is assigned a weight of $w_i = 0.1$. The SMA is formulated as:

$$M_{\text{avg}} = \frac{1}{N} \sum_{i=1}^{N} M_i. \tag{3}$$

The second approach, *Exponential Moving Average (EMA)*, emphasizes later models by assigning weights that decay exponentially, making EMA more sensitive to recent changes. The EMA is expressed recursively as:

$$M_{\text{avg}}^{(i)} = \alpha \cdot M_i + (1 - \alpha) \cdot M_{\text{avg}}^{(i-1)}, \, i \in [2, N], \tag{4}$$

Here, $\alpha$, the smoothing factor (typically between 0 and 1), controls the balance between the current model $M_i$ and the previous EMA result $M_{\text{avg}}^{(i-1)}$.

The third approach, *Weighted Moving Average (WMA)*, also prioritizes recent models but uses a distinct weighting scheme. In WMA, each model is assigned a specific weight, often increasing linearly for later models (e.g., $w_i = i$). The weighted sum is then normalized to compute the average, formulated as follows:

$$M_{\text{avg}} = \sum_{i=1}^{N} \frac{w_i}{w_{\text{sum}}} M_i, \quad w_{\text{sum}} = \sum_{i=1}^{N} w_i. \tag{5}$$

These methods offer flexible ways to combine models based on their recency and relevance. Choosing the right approach depends on the specific application and desired emphasis on newer data.

# 4 Experiments

In this section, we delve into the experimental core of our study, systematically addressing six critical questions surrounding model merging in the context of pre-training: 1) How does model merging

affect performance? 2) How do different merging methods affect final performance? 3) How to determine the optimal interval and number of weights to merge for various model sizes? 4) Do merged pre-trained models contribute to better downstream training? 5) Does model merging improve the stability of training? 6) What processes unfold during model merging? Through these experiments, we aim to provide comprehensive insights into model merging, offering practical guidance for its application and shedding light on its theoretical underpinnings.

## 4.1 How does model merging affect model performance?

Current learning rate schedule methods mainly involve constant learning rates or cosine annealing. In our model pre-training, we employed the Warmup-Stable-Decay (WSD) strategy Hu et al. [2024], which combines a constant learning rate phase with a subsequent cosine decay phase Loshchilov and Hutter [2017]. To explore the effects of model merging under different learning rate schedules, we conducted experiments during both constant learning rate phase and cosine dacay phase.

In the constant learning rate phase, we merged fully trained models of various sizes. As shown in Figure 1, the merged models exhibited significant performance improvements across multiple downstream tasks. For example, on the Humaneval benchmark, Seed-MoE-1.3B/13B improved from 31.1 to 36.6 points, and Seed-MoE-10B/100B increased from 54.3 to 61.6 points. While larger models showed less pronounced gains on certain benchmarks, such as BBH, this was likely due to the near-saturation of these metrics. Overall, the improvements were robust and consistent across model sizes.

Next, we performed model merging in the cosine annealing phase by collecting weights from the annealing stages of Seed-MoE-1.3B/13B, Seed-MoE-10B/100B, and Seed-MoE-15B/150B. As depicted in Figure 2, as the learning rate gradually decreased, the models converged steadily, with performance continuing to improve. Interestingly, at the early annealing stage, the results of `PMA` were comparable to those at the end of the annealing process. In some cases, particularly for larger models, the merged models even surpassed those naturally annealed.

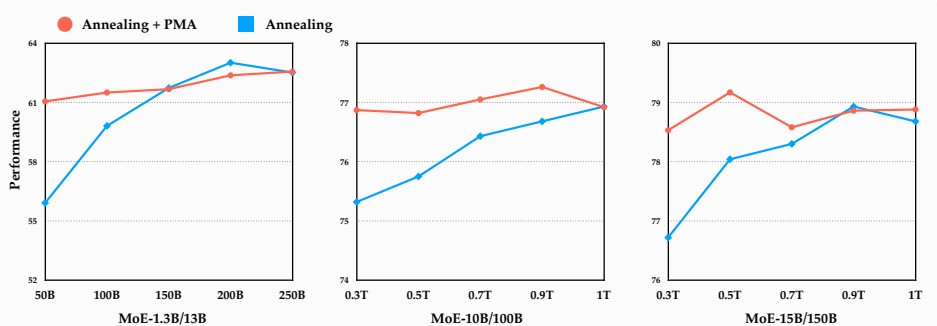

Figure 2: Comparison of overall performance for MoE models of varying sizes under annealing training, before and after model merging. The learning rate follows a cosine schedule during the annealing process. The x-axis shows the count of training tokens.

These findings raised a question: could we simplify the training process by using only the Warmup-Stable phases alongside `PMA`, skipping the decay phase, and avoiding learning rate adjustments? To investigate, we forked two training runs from the stable phase of Seed-MoE-1.3B/13B at 1.4T tokens. One continued with a constant learning rate, while another underwent annealing, each training for an additional 250B tokens. We then merged the models trained with the constant learning rate. As shown in Figure 3, early in training, the merged models significantly outperformed both the constant learning rate and annealed models. Even later, their performance was comparable to the annealed models.

This suggests that **pre-training with a constant learning rate, combined with model merging, can effectively match the performance of an annealed model at any point in the training process without the need for learning rate annealing**. This approach accelerates model validation and significantly reduces computational resource demands.

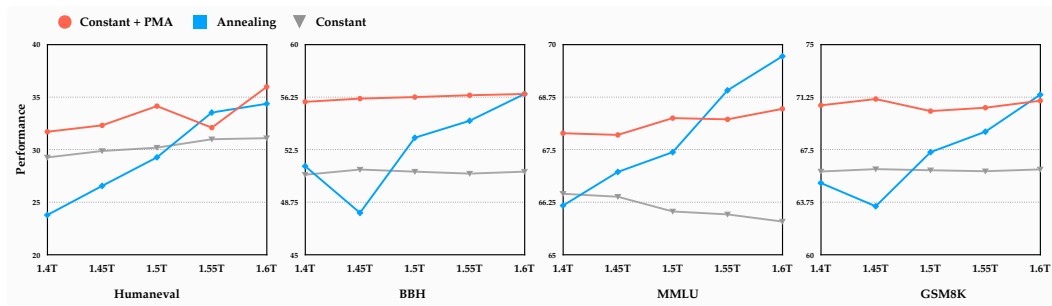

Figure 3: Comparison of downstream task performance between model merging results under stable training and the real annealed model. The x-axis shows the count of training tokens.

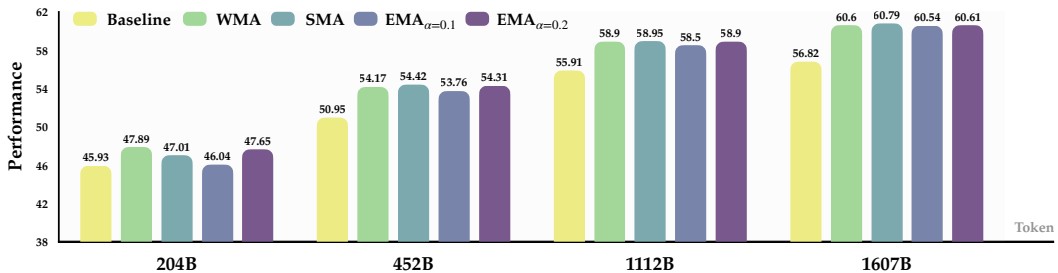

Figure 4: Impact of different model merging methods on final model performance.

## 4.2 How do different merging methods affect final performance?

In this section, we systematically evaluate how different merging strategies affect the performance of merged models. Specifically, we focus on three distinct approaches: EMA, WMA, and SMA. The EMA method employs exponentially decaying weights $w_i = \alpha(1-\alpha)^{N-i}$, giving higher importance to more recent checkpoints. WMA assigns linearly increasing weights $w_i = i$, also prioritizing more recent checkpoints. In contrast, SMA applies uniform weighting, treating all checkpoints equally regardless of their position in the training sequence.

We conducted experiments on Seed-MoE-1.3/13B and showed the results in Figure 4. At 204B training tokens, all merging methods enhanced model performance compared to the pre-merged model, but WMA delivered the best results. This suggests that in the early phases of training, when model weights undergo significant changes, assigning higher weights to checkpoints with more training tokens produces superior models. This is further supported by the fact that $\text{EMA}_{\alpha=0.2}$ outperforms $\text{EMA}_{\alpha=0.1}$. However, as training advances to later stages and model weights stabilize, the performance differences between merging methods diminish. **For its simplicity and stability, we primarily use SMA for model merging in subsequent experiments.**

## 4.3 How to determine the optimal interval and number of weights to merge for various model sizes?

Beyond the merging technique itself, two other factors may also affect the effectiveness of model merging: the interval $V$ between selected models and the number of models $N$. We performed ablation studies on the Seed-MoE-1.3/13B model to investigate these effects, starting with the impact of the interval. As illustrated in the upper part of Figure 5, we fixed $N = 10$ and tested intervals of $V = 4B$, 8B, 16B, and 32B. Notably, at 204B with $V = 32B$, we reduced $N$ to 6 due to insufficient models. In the early stage of training, at 204B tokens, merged results with $V = 16B$ and $V = 32B$ underperformed the baseline. This is likely because large intervals incorporated unstable weights from the initial training phase, leading to significant weight disparities and suboptimal outcomes. As training progressed and weights stabilized, the performance gap across different $V$ settings gradually narrowed. **In practice, the optimal interval scales with model size, following these observed patterns: an interval of around 8B tokens for 1.3B/13B models, 4B tokens for 0.7B/7B models,**

**and approximately 80B tokens for 10B/100B models. This aligns with the tendency of larger models to use larger batch sizes** McCandlish et al. [2018].

Next, we set $V = 8B$ and explored how the number of merged models $N$ affects performance, testing $N = 3, 6, 10$, and 15. As shown in the lower part of Figure 5, early in training, incorporating more models introduced unstable weights, which reduced the performance of merged models. However, once training was complete, merging a larger number of models led to significant performance improvements. Notably, the overall performance for $N = 3$ was nearly 1 point lower than for $N = 15$. To strike a balance between computational cost and performance gains, we opted for $N = 10$ in further experiments.

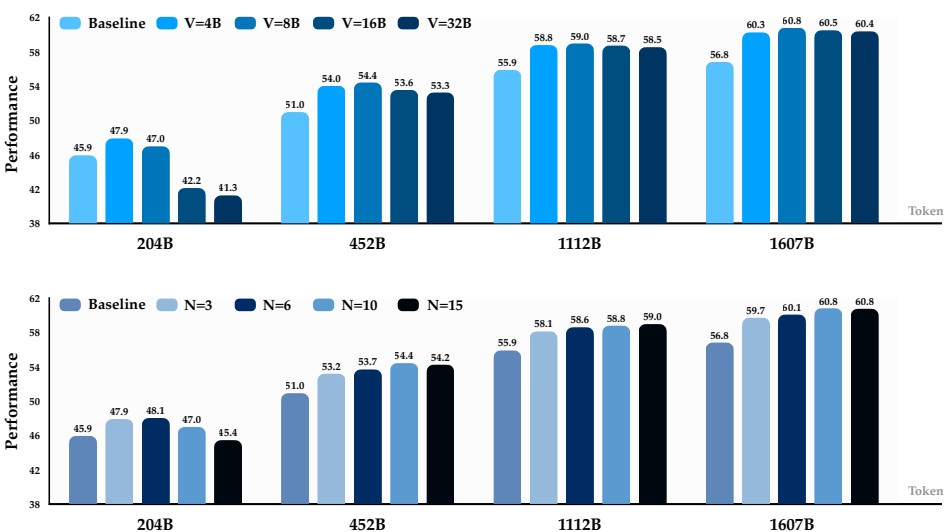

Figure 5: Impact of different model merging hyper-parameters on final model performance of the Seed-MoE-1.3/13B Model.

## 4.4 Do merged pre-trained models contribute to better downstream training?

A complete LLM training process typically involves multiple stages, which are pretraining, continual training (CT), supervised fine-tuning (SFT) and reinforcement learning (RL) in sequence. In light of the capacity of `PMA` to improve pretraining performance, we conjecture that merged pretrained models may similarly prove beneficial for downstream stages. To verify this hypothesis, we initialized downstream training with `PMA`, which we dubbed as `PMA-init`, and investigated its impacts over the baselines (which are initialized from their original checkpoints) for both CT and SFT stages.

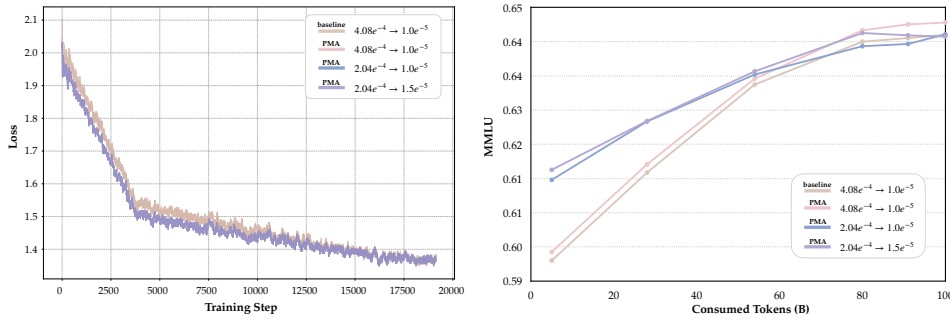

Figure 6: Comparisons of loss curves (left) and performance metrics (right) during CT stage with varying $lr$ schedules, where a cosine scheduler is adopted to decay learning rate from $lr_{peak}$ to $lr_{end}$ (denoted as $lr_{peak} \rightarrow lr_{end}$). `PMA` and `baseline`, stand for whether our `PMA-init` technique is employed or not, respectively.

**CT stage.** We first conducted an ablation study to assess the sensitivity of the `PMA-init` of the CT stage with varying learning rate schedules. Specifically, we experimented with Seed-MoE-0.7B/7B models merged after stable training on approximately 1 trillion tokens. As illustrated in Figure 6 (left), the initialization weights obtained via PMA consistently achieved marginally lower loss at the initial training phase, against the baseline with the same training configuration. As training progresses, the loss values for models with different initialization weights converge to comparable levels. It's worth noting that in the loss curve, the purple line significantly overlaps with the blue line, and the brown line significantly overlaps with the pink line. Another observation is made in the Figure 6 (right), where evaluation on the MMLU benchmark reveals that the `PMA-init` models outperform the baseline early in training. While these models tend to retain a slight performance edge in later stages, their results on other tasks may be slightly suboptimal, leading to overall performance parity with the baseline. Experiments across varied learning rate schedules corroborate these findings, indicating that **models converge to similar performance levels by the end of training, and no extensive learning rate tuning is required for** `PMA-init`.

**SFT stage.** We next analyzed the impact of `PMA-init` on the SFT stage, where the detailed results can be found in the Appendix B. Although initialization with merged weights occasionally yields performance improvements, such gains are not consistently observed. Nonetheless, this approach does not adversely affect downstream training outcomes and may be a viable strategy for researchers seeking to enhance model performance.

### 4.5 Does model merging improve the training stability?

In large-scale LLM training, infrastructure issues are almost inevitable and often lead to training instability phenomena such as loss spikes or diverging. Specifically, a loss spike occurs when, at a specific point during the multi-stage training, the model's predictions deteriorate significantly compared to previous iterations. This phenomenon is often observed alongside gradient norm (GradNorm) explosion during backpropagation, which causes large weight updates and eventually lead to a irrecoverable spike in its loss function Cohen et al. [2021]. In the experiments detailed in Section 4.4, as illustrated in Figure 7 (left), we observed that a model initialized with `PMA-init` for SFT stage demonstrated a notably more stable GradNorm metric compared to the baseline. This stability is also evident in the reduced frequency of loss spikes relative to the baseline. Since applying `PMA-init` for downstream training does not impact the model's final performance and remains robust across different learning rates, we established a series of experiments to explore whether model merging could enhance training stability.

Given the extremely high expenses associated, it is unfeasible to conduct a direct analysis of training instability in LLM pre-training. Experiments Wortsman et al. [2024] show that small models using a relatively large learning rate will exhibit unstable training characteristics similar to those of large models. We thus reproduce the instability phenomena on small models to study the influence of our `PMA-init` on training stability. In one such experiment, we trained a 330M/3.3B MoE model from scratch using an exceptionally high learning rate of 6e-3. As shown in Figure 7 (right), the model overshot the optimal weights, resulting in unstable training and abrupt loss spikes as expected, and was irreversible to its original trajectory. To address this, we adopted `PMA-init` with three checkpoints saved before the training collapse happened, to resume the pre-training process. As depicted by the red line in Figure 7 (right), the resumed training process stabilized, successfully navigating past the point of the loss spike and continuing along its original training trajectory.

**These results highlight that** `PMA-init` **can reliably enhance the multi-stage training stability.** When a loss spike occurs, one can merge the model checkpoints from before the spike and resume training from that point. This approach provides an alternative solution to avoid retraining the model from scratch, thereby substantially reducing the waste of computational resources.

### 4.6 Investigating the Mechanisms of Model Merging

To gain deeper insight into the underlying mechanisms that enable model merging to be effective, we provide both qualitative and quantitative analyses, employing mathematical derivations and visualizations of weight distributions. Due to space limitations, this section focuses solely on the qualitative analysis. For the quantitative analysis, please refer to Appendix C for more details.

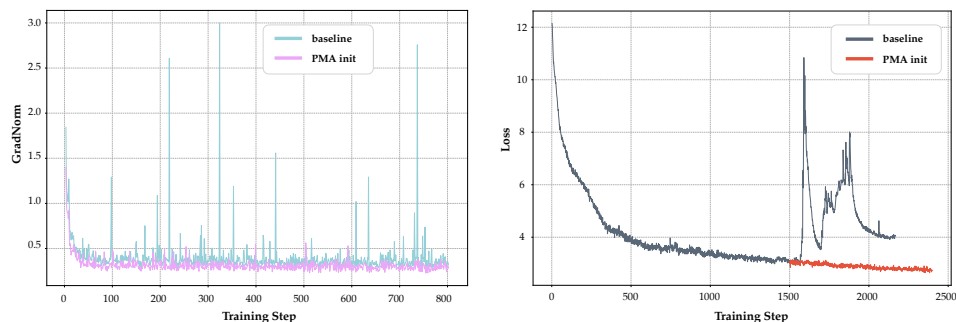

Figure 7: Left: GradNorm comparisons for SFT training initialized with `PMA-init`. Right: Comparison of pre-training loss curves between resuming with `PMA-init` and the original training.

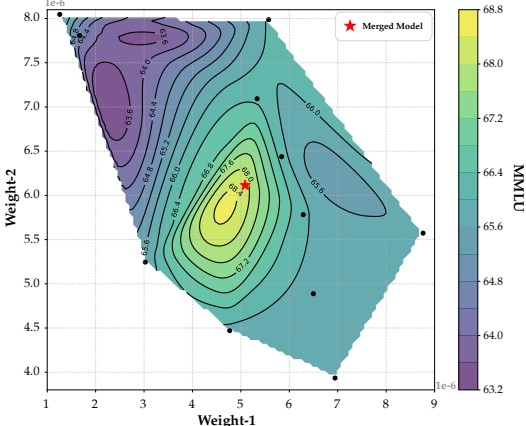

Figure 8: Visualization of MMLU score contour lines, comparing the weights of an original model with those of a merged model. Black dots represent the parameter locations of various individual model checkpoints.

We selected several checkpoints from the pre-training of Seed-MoE-1.3B/13B and visualized the average distribution of two selected parameters from a specific layer. Using these points, we generated contour lines for MMLU scores, as illustrated in Figure 8. The weight positions of various individual models are marked as black dots. These dots are distributed along the MMLU score contours, revealing a discernible "complementary" pattern. The averaged weight position (representative of the merged model) is often situated closer to a region of higher MMLU scores (a better optimum) than many individual model checkpoints. This visualization also provides an intuitive explanation for why model merging yields diminished improvements when models are annealed to a very low learning rate: at such a stage, the models to be merged are already tightly converged within a specific local optimum. Merging them essentially averages points within this already narrow basin, making it unlikely to escape to a significantly better or different optimal region.

## 5 Conclusion

This research pioneers a deeper exploration of model merging within the challenging pre-training stage of large-scale models. By training a spectrum of MoE and Dense models and performing rigorous ablations, we established that merging checkpoints from stable training phases not only yields significant performance gains and predicts annealing but also streamlines development and reduces costs. Our work provides concrete guidance on merging strategies, optimal parameters, and downstream applications, alongside insights into the underlying mechanisms. These contributions equip the open-source community with the knowledge and tools for more efficient model development through pre-training merging.

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

## A   The Effect of Model Merging in Dense Models

We also conducted model merging experiments on Dense architecture models, which mirror the exact architecture of LLaMA 3.1 [Grattafiori et al., 2024], ranging from small Dense-411M models to large Dense-70B models. Since the 411M and 2B models were not sufficiently trained, we used a configuration of N=6 for merging, with weight intervals (V) of 2B and 5B tokens, respectively. For the 8B and 70B models, which were trained more thoroughly, we used N=10, with V values of 15B and 40B for merging. As shown in Figure 9, models of different sizes achieved significant improvements on downstream tasks after model merging. Notably, the performance gains of larger models were not smaller than those of smaller models. Specifically, Dense-70B improved from 50.6 to 57.9 on Humaneval and from 85.9 to 91.3 on GSM8K. This further validates the robustness and generalization ability of PMA, demonstrating that it can work across different model architectures and sizes.

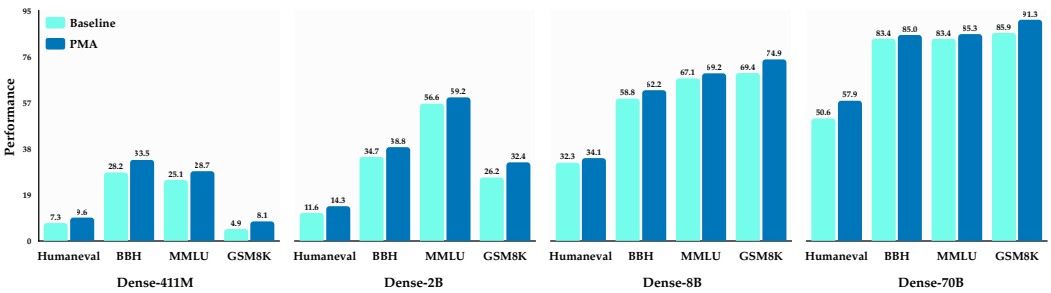

Figure 9: Comparison of downstream task performance for dense models of varying sizes under stable training, before and after model merging.

## B   Model Merging at the CT Stage for Supervised Fine-Tuning

PMA-init can be integrated during the CPT annealing phase, where PMA weights are merged to provide a strong starting point for fine-tuning. We conducted an ablation study to assess the sensitivity of the PMA-init during the SFT stage to varying learning rate schedules. This study included experiments on merged MoE-15B/150B models following stable training on approximately 16T tokens, as well as after further training on 1T tokens with cosine annealing. We conducted SFT training for 220M tokens using both the original weights and PMA-init weights. For the original weights, we used a cosine learning rate schedule with an initial learning rate of 2e-5 and an end learning rate of 2e-6. For the PMA-init weights, we used cosine schedules with initial learning rates of 1e-5, 2e-5, and 4e-5, all with an end learning rate of 2e-6. We evaluated the trained models using Open-Benchmark, which includes MMLU Hendrycks et al. [2021], LiveBench White et al. [2025], AMC-2023, GPQA Rein et al. [2024] and LiveCodeBench White et al. [2025], as well as our in-house evaluation set comprising OOD, Reasoning, and Instruction Following assessments. As shown in Table 1, with the same learning

Table 1: Comparisons of performance metrics during SFT stage with varying $lr$ schedules, where a cosine scheduler is adopted to decay learning rate from $lr_{peak}$ to $lr_{end}$ (denoted as $lr_{peak} \rightarrow lr_{end}$). PMA and baseline, stand for whether our PMA-init technique is employed or not, respectively. IF refers to Instruction Following.

| Model | Open-Benchmark | | | | | In-house Evaluation | | |
| --- | --- | --- | --- | --- | --- | --- | --- | --- |
| | MMLU | LiveBench | AMC-2023 | GPQA | LiveCodeBench | OOD | Reasoning | IF |
| Baseline$_{2e-5 \rightarrow 2e-6}$ | 86.8 | 50.5 | 61.0 | **55.2** | **39.7** | 32.6 | 32.1 | 36.3 |
| PMA$_{2e-5 \rightarrow 2e-6}$ | 87.1 | 52.0 | 64.0 | 54.0 | 39.4 | **34.7** | **34.0** | **38.8** |
| PMA$_{1e-5 \rightarrow 2e-6}$ | **87.2** | **53.2** | **65.5** | 54.4 | **39.7** | 33.8 | 33.2 | 37.3 |
| PMA$_{4e-5 \rightarrow 2e-6}$ | 87.0 | 51.3 | 61.4 | 54.0 | 39.2 | 31.8 | 32.6 | 37.2 |

rate, PMA-init significantly outperformed the baseline on both Open-Benchmark and our in-house evaluations. Notably, on the in-house evaluation set, we observed improvements of over two points in OOD and Instruction Following, and a 1.9-point increase in Reasoning. In the other two experiments

with different learning rates, we also saw some degree of improvement compared to the baseline, especially with $\mathtt{PMA}_{1e^{-5}\to 2e^{-6}}$, which showed gains of 2.7 points on Livebench and 4.5 points on AMC-2023.

However, we were unable to replicate such significant gains in subsequent experiments with other model sizes, although it did not negatively impact the final downstream model performance. Therefore, as a low-cost approach, $\mathtt{PMA}$-init is worth trying to obtain a more powerful downstream model.

## C   Quantitative Analysis on Model Merging

We begin with a second-order Taylor expansion of the loss function $L(\theta)$ around an optimal parameter set $\theta^*$:

$$L(\theta) \approx L(\theta^*) + (\theta - \theta^*)^T \nabla L(\theta^*) + \frac{1}{2}(\theta - \theta^*)^T H(\theta - \theta^*), \tag{6}$$

where $H$ is the Hessian matrix of the loss function evaluated at $\theta^*$ (the matrix of second partial derivatives), which captures curvature information. Since $\theta^*$ is an optimal point, the gradient $\nabla L(\theta^*)$ is zero. Thus, the expansion simplifies to:

$$L(\theta) \approx L(\theta^*) + \frac{1}{2}(\theta - \theta^*)^T H(\theta - \theta^*). \tag{7}$$

Consider $k$ sets of model parameters $\theta_1, \theta_2, \ldots, \theta_k$. Let the deviation vector of each model $i$ from the optimal parameters be $\delta_i = \theta_i - \theta^*$. The loss for each model $i$ can then be approximated as:

$$L(\theta_i) \approx L(\theta^*) + \frac{1}{2}\delta_i^T H \delta_i. \tag{8}$$

The average loss of these $k$ individual models is:

$$\frac{1}{k}\sum_{i=1}^{k} L(\theta_i) \approx L(\theta^*) + \frac{1}{2k}\sum_{i=1}^{k} \delta_i^T H \delta_i. \tag{9}$$

The parameters of the merged model are $\theta_{\text{avg}} = \frac{1}{k}\sum_{i=1}^{k}\theta_i$. The deviation of this merged model from the optimal parameters is $\theta_{\text{avg}} - \theta^* = \frac{1}{k}\sum_{i=1}^{k}\delta_i$. The loss for the merged model is approximated by:

$$L(\theta_{\text{avg}}) \approx L(\theta^*) + \frac{1}{2}\left(\frac{1}{k}\sum_{i=1}^{k}\delta_i\right)^T H \left(\frac{1}{k}\sum_{i=1}^{k}\delta_i\right). \tag{10}$$

Expanding the quadratic term:

$$\frac{1}{2}\left(\frac{1}{k}\sum_{i=1}^{k}\delta_i\right)^T H \left(\frac{1}{k}\sum_{i=1}^{k}\delta_i\right) = \frac{1}{2k^2}\sum_{i=1}^{k}\sum_{j=1}^{k}\delta_i^T H \delta_j. \tag{11}$$

This can be rewritten by separating diagonal and off-diagonal terms:

$$\frac{1}{2k^2}\left(\sum_{i=1}^{k}\delta_i^T H \delta_i + \sum_{i=1}^{k}\sum_{j\neq i}\delta_i^T H \delta_j\right). \tag{12}$$

For the merged model to have a lower loss than the average loss of the individual models, i.e., $L(\theta_{\text{avg}}) < \frac{1}{k}\sum_{i=1}^{k} L(\theta_i)$, the following condition must hold:

$$\frac{1}{2k^2}\left(\sum_{i=1}^{k}\delta_i^T H \delta_i + \sum_{i=1}^{k}\sum_{j\neq i}\delta_i^T H \delta_j\right) < \frac{1}{2k}\sum_{i=1}^{k}\delta_i^T H \delta_i. \tag{13}$$

Multiplying by $2k^2$ and rearranging terms, we get:

$$\sum_{i=1}^{k}\delta_i^T H \delta_i + \sum_{i=1}^{k}\sum_{j\neq i}\delta_i^T H \delta_j < k\sum_{i=1}^{k}\delta_i^T H \delta_i. \tag{14}$$

Which simplifies to:

$$\sum_{i=1}^{k} \sum_{j \neq i} \delta_i^T H \delta_j < (k-1) \sum_{i=1}^{k} \delta_i^T H \delta_i. \tag{15}$$

Assuming $H$ is a positive definite matrix (which is generally true around a local minimum), then each term $\delta_i^T H \delta_i > 0$. The inequality is more easily satisfied if the off-diagonal terms $\delta_i^T H \delta_j$ (for $i \neq j$) are predominantly negative. This "negative correlation" in the context of the Hessian means that the deviation vectors point in somewhat opposing directions relative to the curvature of the loss landscape. This mathematical analysis can be intuitively interpreted as follows: 1. The effectiveness of model weight merging stems from the fact that different model checkpoints, representing different points in the training trajectory, have explored different local regions or directions within the parameter space. 2. When these explorations exhibit a degree of "complementarity" concerning the geometric structure of the loss function (captured by the Hessian and the cross-terms $\delta_i^T H \delta_j$), their average can position the merged model closer to an optimal point than the individual models might be on average. 3. This helps explain why merging models, particularly those from a stable yet ongoing training phase, often improves performance. The averaging process can smooth out idiosyncrasies of individual checkpoints. This analysis suggests that weight merging is not merely a simple averaging of parameters but rather a process that can leverage the geometric structure of the loss landscape and the diversity among the models being merged.

## D   Limitations

In our study, we thoroughly investigated the potential of model merging in the pre-training phase, offering significant advantages for teams working on large-scale model pre-training to pursue more daring explorations. This is due to the fact that model merging can replicate the benefits of simulated annealing, greatly shortening the exploration period during pre-training. While our experiments were extensive, certain aspects still remain open for deeper research.

In our experiments, we defaulted to using the optimal learning rate derived from the scaling law for model training, without extensively exploring the impact of learning rate on model merging. In our practice, we believe that training with a higher learning rate could lead to a better model through model merging, which aligns with the findings in Sanyal et al. [2024]. However, due to the high computational cost, we did not further quantify the impact of learning rate on model merging in a more detailed manner.

Additionally, this paper primarily focuses on the application of model merging in pre-training. In reality, due to innovations in RL algorithms [Yu et al., 2025, Yuan et al., 2025, Shao et al., 2024], RL training has become more stable and often involves longer training cycles, during which a series of adjacent weights can be obtained. This paper does not investigate model merging in the context of post-training scenarios, and we leave this aspect for future research.

## E   Societal Impact

The development of large-scale language models, while technologically promising, carries significant societal implications that necessitate careful consideration. Our research on model merging during pre-training offers advancements that not only push the technical frontier but also contribute positively to the responsible development of artificial intelligence.

### E.1   Potential Benefits

The primary contribution of our work is a substantial increase in the efficiency of the LLM pre-training process. The Pre-trained Model Averaging (PMA) technique significantly reduces the computational resources and time required to develop high-performance models. This efficiency has several downstream societal benefits:

**Accelerating Innovation and Accessibility:** By lowering the immense costs traditionally associated with training state-of-the-art models, our methods can help democratize access to large-scale AI development. This allows smaller research labs, academic institutions, and companies to contribute to the field, fostering broader innovation.

**Enhancing Applications:** The accelerated development cycles enabled by our work can lead to faster improvements in critical NLP applications. This includes creating more effective and accessible educational tools, developing more nuanced and accurate communication technologies (such as real-time translation), and building more capable assistive technologies for individuals with disabilities.

## E.2 Risks and Mitigation Strategies

We also recognize the potential risks associated with large-scale models and believe our work provides avenues for their mitigation.

**Computational Energy Costs and Environmental Impact:** The training of LLMs is an energy-intensive process with a considerable environmental footprint. Our findings directly address this concern. By demonstrating that merged checkpoints can match or exceed the performance of models that have undergone a full, lengthy annealing phase, our technique provides a method to curtail total training time. Furthermore, the ability of PMA-init to recover from training instabilities prevents the wasteful process of restarting training from scratch. These efficiencies translate directly into reduced energy consumption and a smaller carbon footprint for model development.

**Algorithmic Bias:** Large language models can perpetuate and amplify existing societal biases present in their training data. While our research does not introduce a new method for bias detection, the efficiency it creates is a critical enabler for more rigorous and responsible AI development. By reducing the cost of each training and evaluation cycle, our methods allow development teams to:

**Iterate More Frequently:** Conduct more frequent and thorough testing for biases within the model.

**Reallocate Resources:** Dedicate computational and financial resources saved from training towards crucial mitigation tasks, such as curating higher-quality, more diverse datasets and implementing sophisticated fairness-aware fine-tuning techniques.

In summary, the model merging techniques presented in this paper offer a practical path toward a more efficient, sustainable, and responsible ecosystem for AI development. By addressing the core challenge of computational cost, we empower the research community to not only build more powerful models but also to invest more deeply in making them safe and beneficial for society.

