# OpenReview forum: "Model Merging in Pre-training of Large Language Models"
_NeurIPS.cc/2025/Conference — NeurIPS 2025 poster_

### Official Review · Reviewer_DRRf · 2025-06-16

**Clarity:** 3
**Significance:** 4
**Originality:** 1
**Rating:** 4
**Confidence:** 4

**Summary:**

The paper investigates the utility of merging multiple consecutive checkpoints at the pre-training stage of model development. Paper includes experiments with MoEs and dense models (in the appendix). Among other things, authors find that model merging can significantly improve the downstream performance of the models, make CPT more stable, as well make final learning rate annealing unnecessary.

**Questions:**

- authors claim that PMA is a novel framework (l.71), but wasn't a very similar strategy already been applied in e.g. LAWA? Isn't this just application of the LAWA strategy at scale?
- Authors say: "Through extensive experiments, we demonstrate the effectiveness of PMA-init on both CT and SFT stages." This sound somewhat misleading to me, i.e. for CT authors show more stable training but no clear performance advantages, for SFT the gains in performance are also very limited and not consistent as acknowledged by the authors;
- why is the main focus on MoEs and not dense models? Since the paper includes dense mdoels merging in appendix, it would be  nice to include a comparison of the effectiveness of PMA in MoEs vs Dense

Further limitations & comments:
- related works is missing an important work [1]
- authors did not include more sophisticated weight merging methods such as TIES, Fisher merging etc.
- authors mention significant computation saving of merging but do not quantify the saving
- Section 4.3 is titled "How to determine the optimal interval and number of weights to merge for various model
sizes?", but the content of the section is not focusing on how to efficiently determine those parameters, but rather on descriptive analysis of the said hyperparameters
- In the NeurIPS Paper Checklist authors state that experimental results are reproducible, which is not the case given that the model architecture has not been publicly released and is not described in the detail in the paper (ll.130-131).

[1] Yadav, Prateek, et al. "What Matters for Model Merging at Scale?." arXiv preprint arXiv:2410.03617 (2024)

**Ethical Concerns:**

["NO or VERY MINOR ethics concerns only"]

**Final Justification:**

While I definitely acknowledge the practical relevance of this work, I am concerned by the novelty I acknowledge that scaling existing techniques is challenging in practice) as well as limited reproducibility. Hence I keep my "weak accept" assessment.

**Limitations:**

This paper includes experiments with models of very large scale (dense of up to 70-B) but provides no discussion on potential societal impact.
The paper does not provide any details on the data that has been used for training the models, no information about copyright, privacy etc. is provided, which is concerning.

**Paper Formatting Concerns:**

-

**Quality:**

3

**Strengths And Weaknesses:**

Overall, the paper makes a valuable contribution by empirically validating the effectiveness of known model merging techniques for pre-training modern LLMs using contemporary pre-training pipelines. I believe the results presented are valuable to the community and practitioners, especially given the substantial compute expenditure on the presented experiments. I found particularly interesting the insight that merging the k latest checkpoints can achieve similar performance gains as complete learning rate annealing. Additionally, the paper is well written, uses simple language and is easy to follow.

That being said, the main weakness of the paper is its lack of methodological novelty; while empirical validation of these insights is valuable, many insights are somewhat intuitive (merging well converged models is better, more checkpoints etc.).

Overall, while clearly valuable for the practitioners, I am not 100% sure that empirical validation of existing techniques at scale constitute sufficient contribution to NeuIPs.

---

> ### Author Rebuttal · Authors · 2025-07-31
>
> We appreciate the reviewer’s detailed evaluation and valuable feedback on our work. I hope my responses below effectively address your concerns.
> ### **Weakness 1 : the main weakness of the paper is its lack of methodological novelty; while empirical validation of these insights is valuable, many insights are somewhat intuitive (merging well converged models is better, more checkpoints etc.). Overall, while clearly valuable for the practitioners, I am not 100% sure that empirical validation of existing techniques at scale constitute sufficient contribution to NeuIPs.**
> ### **Answer W1:**
>   We sincerely thank the reviewer for their thoughtful feedback and for recognizing the clear value of our work for practitioners. We address the main concern regarding methodological novelty and the perceived intuitiveness of some insights below, clarifying why our contributions are significant and well-suited for NeurIPS.
>
> We acknowledge that **our paper prioritizes simplicity over introducing complex, novel techniques. This was an intentional design choice.** By focusing on straightforward model-level merging methods without intricate module-level tricks, we aimed to develop a generalizable approach that is robust across diverse scenarios and model architectures. Complex techniques often risk overfitting to specific settings, limiting their practical applicability. In contrast, our simple yet effective method demonstrates broad utility, which we believe is a key strength. The empirical validation of such simplicity at scale is non-trivial and provides actionable insights for the community, particularly as large-scale models become increasingly prevalent.
>
>   The reviewer notes that some insights, such as "merging more checkpoints is better," may seem intuitive. While this observation may appear straightforward in hindsight, our experiments reveal important nuances. **For instance, merging more checkpoints often implies larger temporal gaps between checkpoints, which can introduce unforeseen challenges, such as compatibility issues or degraded performance in certain tasks.** Our work rigorously quantifies these trade-offs, providing empirical evidence to validate or challenge intuition. As reviewer says, they offer actionable guidance for practitioners.
>
>   Moreover, **scaling insights from smaller models to larger ones is far from trivial.** Our work validates these insights at scale, showing that principles effective for smaller models hold for larger, more complex models and datasets. This scalability is not guaranteed, and our comprehensive experiments provide valuable evidence of robustness, making our results highly insightful for the field. **Meanwhile, we have also done extensive explorations in areas such as applying model merging to downstream training and addressing stability issues. To the best of my knowledge, no previous studies have conducted such explorations.**
>
>   We believe our contribution aligns with NeurIPS’ mission to advance AI by disseminating impactful, practical knowledge. While methodological novelty is valuable, empirical validation of scalable, generalizable techniques is equally critical for driving real-world progress in large language model development. **Our work delivers actionable, reproducible insights that benefit the broader AI community.** We hope NeurIPS will serve as a platform to amplify these findings, fostering further innovation and adoption in the field.
>
> ### **Question1: authors claim that PMA is a novel framework (l.71), but wasn't a very similar strategy already been applied in e.g. LAWA? Isn't this just application of the LAWA strategy at scale?**
> ### **Answer Q1:**
> We acknowledge that our description on line 71 may have caused some ambiguity. To clarify, our intent was not to claim that PMA introduces entirely new methodological concepts but rather to emphasize its novel application and design as a streamlined, scalable framework for model merging. LAWA demonstrated that merging checkpoints at intermediate stages can significantly accelerate training. In this paper, **we not only explore scaling further, which is a non-trivial endeavor, but also conduct experiments on additional merging details, such as the interval between merges. Additionally, we perform extensive explorations in areas such as applying model merging to downstream training and addressing stability issues. Overall, we believe our paper differs significantly from LAWA in multiple aspects.**
>
> ### **Question2 : "Through extensive experiments, we demonstrate the effectiveness of PMA-init on both CT and SFT stages." This sound somewhat misleading to me, i.e. for CT authors show more stable training but no clear performance advantages, for SFT the gains in performance are also very limited and not consistent as acknowledged by the authors;**
> ### **Answer Q2:**
>   We acknowledge that the phrasing could be clearer and offer the following clarifications:
>  -  PMA-init achieves significant improvements of ~2 points on challenging OOD and Reasoning tasks in SFT, which are **substantial given the difficulty of these benchmarks.** While gains vary across tasks, these results highlight PMA-init’s effectiveness.
>  - Generally, **PMA-init enhances training stability without performance loss, enabling more reliable training and occasionally yielding stronger models, demonstrating its practical value.**
> ### **Question3: Why is the main focus on MoEs and not dense models? it would be nice to include a comparison of the effectiveness of PMA in MoEs vs Dense**
> ### **Answer Q3:**
>   The primary focus on MoE architectures in our paper is motivated by their increasing prominence in recent advancements in large-scale language models. Over the past six months, **MoE has emerged as the dominant architecture for flagship models due to its scalability, computational efficiency, and strong performance in diverse tasks (e.g., like Deepseek-V3, Qwen3 and others).** Given this trend, we believe that studying PMA in the context of MoEs aligns with the future direction of large model development, making it a timely and impactful contribution.
>
>   That said, we fully recognize the importance of dense models, which remain widely used and relevant. To ensure a comprehensive evaluation, we included experiments on dense model merging in Appendix A. **These results demonstrate that PMA achieves consistent effectiveness across both MoE and dense architectures, highlighting its versatility.**
>
> ### **Question 4:related works is missing  [1]**
> ### **Answer Q4:**
>   Thank you for pointing out the omission of the important work [1] in our related works section. We sincerely apologize for this oversight.  In the revised version of the manuscript, we will incorporate a discussion of [1] in the related works section, highlighting its contributions and relevance to our research.
> [1] Yadav, Prateek, et al. "What Matters for Model Merging at Scale?." arXiv preprint arXiv:2410.03617 (2024)
>
> ### **Question5: did not include more sophisticated weight merging methods**
> ### **Answer Q5:**
>   We acknowledge that our paper prioritizes simplicity over introducing complex, novel techniques. Refer to **Answer W1**.
>
> ### **Question 6:authors mention significant computation saving of merging but do not quantify the saving**
> ### **Answer Q6:**
> In the manuscript, we reference Figure 2, which illustrates that during the CT phase, our proposed PMA approach achieves performance comparable to full annealing while training on only 50% of the token volume. **This effectively translates to a 50% reduction in computational cost for the CT phase compared to the baseline full annealing method.** We believe this constitutes a significant saving in computational resources.
>
> ### **Question7: Section 4.3 is not focusing on how to efficiently determine those parameters, but rather on descriptive analysis of the said hyperparameters**
> ### **Answer Q7:**
>  In practical model training, it is challenging to precisely determine the optimal merging hyperparameters for specific model sizes. This is because, in large-scale model training, it is infeasible to densely save checkpoints for merging due to computational and storage constraints. Therefore, we can only provide qualitative results. **Generally, larger models require larger merging intervals, and merging around 10 weights is typically sufficient. We believe this level of insight is adequate to provide developers with a solid understanding.**
>
> ### **Question 8:In the NeurIPS Paper Checklist authors state that experimental results are reproducible, which is not the case given that the model architecture has not been publicly released and is not described in the detail in the paper (ll.130-131).**
> ### **Answer Q8:**
>   While we acknowledge that these details were not fully disclosed, our conclusions are designed to be architecture-agnostic. To ensure robustness, we conducted experiments with both MoE and dense models, where the dense model mirrors the exact architecture of LLaMA 3.1. **Our findings hold across both architectures, demonstrating that our contributions are not dependent on specific MoE structural details.**
>
> ### **Question9: This paper includes experiments with models of very large scale (dense of up to 70-B) but provides no discussion on potential societal impact. The paper does not provide any details on the data that has been used for training the models, no information about copyright, privacy etc. is provided, which is concerning.**
> ### **Answer Q9:**
>  We fully agree with the reviewer on the importance of transparency and discussing societal impacts, and we appreciate your pointing out these areas for improvement. Our primary intention is to share technical details to contribute to the academic community, rather than directly promoting specific application scenarios or commercial deployment. **Therefore, we believe the conclusions of the paper are independent of specific data and do not pose significant societal risks.**

---

> > ### Comment · Reviewer_DRRf · 2025-08-04
> >
> > I thank the authors for their replies.
> > While I definitely acknowledge the practical relevance of this work, I will leave my score unchanged, given that the authors are not entirely clear about the societal impact, and that reproducibility is questionable given that model architecture/datasets are not provided.

---

> > > ### Author Response · Authors · 2025-08-07
> > >
> > > Dear Reviewer,
> > >
> > > Thank you for your valuable feedback and for recognizing the practical relevance of our work. We appreciate your concerns regarding the societal impact and reproducibility of our study. Below, we provide detailed responses to your comments for  addressing these points.
> > >
> > > ### On Societal Impact
> > > We fully agree on the importance of discussing societal implications, and we sincerely apologize for not addressing this in the original submission. To rectify this, we will add a new section titled “Societal Impact” (Section 6) in the next revised manuscript. This section will outline the potential benefits of our large-scale models, such as improving efficiency in applications like natural language processing, which can enhance accessibility in education and communication. It will also address risks, including computational energy costs and potential biases, with mitigation strategies such as optimized inference and bias detection frameworks. These additions aim to provide a clear and comprehensive discussion of our work’s societal implications, aligning with responsible AI research practices.
> > >
> > > ### On Reproducibility and Data Transparency
> > > We acknowledge your concerns about reproducibility due to limited details on model architecture and datasets. Our dense model is based on the open-source Llama3 architecture, which is publicly available and widely used.  Regarding datasets, institutional and privacy restrictions prevent us from releasing the primary datasets. However, our conclusions focus on general scaling properties and are independent of specific datasets. To further validate this, we plan to conduct additional experiments using the open-source OLMoE model[1] and publicly available datasets, with results to be included in Appendix E. These experiments will confirm that our conclusions are robust across different data and architectures.
> > >
> > > We believe these revisions fully address your concerns and significantly improve the transparency and reproducibility of our work. We are happy to provide further clarifications or incorporate additional details as needed. Thank you again for your valuable feedback, which has greatly strengthened our manuscript.
> > >
> > > Sincerely
> > > [1] Muennighoff et al. (2024). Olmoe: Open mixture-of-experts language models. arXiv preprint arXiv:2409.02060.

---

> > > > ### Author Response · Authors · 2025-08-08
> > > >
> > > > Dear Reviewer,
> > > >
> > > > We would like to express our sincere appreciation for the time and effort you have dedicated to reviewing our paper.
> > > >
> > > > As the author-reviewer discussion period concludes on August 8th at 11:59 PM AoE, we wanted to gently follow up. We hope our rebuttal has adequately addressed your concerns. Please let us know if you have any further questions. We would be happy to provide additional clarifications.
> > > >
> > > > Best regards,
> > > >
> > > > The Authors

---

### Official Review · Reviewer_fgbv · 2025-07-01

**Clarity:** 3
**Significance:** 3
**Originality:** 2
**Rating:** 4
**Confidence:** 5

**Summary:**

In this paper, authors study model merging in LLM pretraining. Useful insights on selecting hyperparameters for model merging, the effects of model merging on pretraining performance and downstream task performance, the benefit of model merging in helping with training stability are provided through extensive experiments.

**Questions:**

1. Minor: The related work section looks a bit repetitive / is similar in structure with the first few paragraphs of introduction.

2. For figure 2, I did not find the name of the benchmark(s) for which the performance is reported. Perhaps it’s an average performance?
Authors frame the experiment section as tackling one research question for each subsection. It would be nice if there are fonts in bold to summarize the findings in each of subsection (currently only 4.1 does that).

**Ethical Concerns:**

["NO or VERY MINOR ethics concerns only"]

**Final Justification:**

I believe the paper will be strengthened by organizing the setting details, which is particularly important for an empirical work like this. I will keep my current positive rating and encourage authors to include these revisions in the camera-ready version.

Some of my questions are listed as future works, which is fine. I find the empirical exploration of this paper valuable to the community.

**Quality:**

3

**Strengths And Weaknesses:**

Strength:
Model merging is under-explored in pretraining stage. This paper provides useful insights and guidelines through extensive experiments to a reasonable scale.


Weakness/Questions
1. The experiment settings are disorganized. Although I can guess the setting from the context for most of times, I encourage authors to systematically include settings for the experiment section. For example, for PMA-init, is it used with warmup-stable phase or annealing phase?

2.
> pre-training with a constant learning rate, combined with model merging, can200
effectively match the performance of an annealed model at any point in the training process201
without the need for learning rate annealing.

To eliminate the need for learning rate annealing as authors claimed, authors should perform experiments on the comparison between (annealed model  + post-training) and (merging constant learning rate model + post-training). If 4.4 is performed using (merging constant learning rate model), then this claim in 4.1 is valid. Otherwise, there should be more experiments to support the claim quoted.

2. On 262~266, I personally have observed similar behaviors on vision classification model pretraining, where EMA improves pretraining metric while the benefit does not transfer to downstream training. Although this result may come as negative, I think it is valuable information to the community and authors can consider discussing more (and include results) on the results from SFT stage. I also encourage authors to conduct more empirical analysis on the reason why PMA’s benefit does not transfer to downstream finetuning.

4. Although model merging between different finetuned models are well-studied, I don’t see works that evaluate EMA/SMA on the same training trajectory during the post-training stage. I wonder how many of these conclusions in pretraining merging transfer to post-training.

---

> ### Author Rebuttal · Authors · 2025-07-31
>
> We appreciate the reviewer’s detailed evaluation and valuable feedback on our work’s training details. I hope my responses below adequately address your concerns.
> ### **Weakness 1: The experiment settings are disorganized. Although I can guess the setting from the context for most of times, I encourage authors to systematically include settings for the experiment section. For example, for PMA-init, is it used with warmup-stable phase or annealing phase?**
> ### **Answer W1:**
> We fully agree that clearly documenting the settings for the experiment section enhances reproducibility and clarity, and we appreciate your encouragement to improve this aspect of our work. Regarding the specific question about PMA-init, we would like to clarify its usage across different training phases. PMA-init can be applied flexibly depending on the training objective:
>
>   **1. Warmup-Stable Phase:** If the goal is to resume training and enhance model stability, PMA-init can be employed during the warmup-stable phase to initialize the model with robust weights, thereby improving training convergence and stability.
>
>   **2. CPT (Continued Pre-Training) with Annealing Phase:** When used in the context of continued pre-training (CPT), PMA-init can be applied during the stable phase to provide initialized weights for downstream CPT annealing, ensuring a smoother transition to fine-tuning.
>
>   **3. SFT (Supervised Fine-Tuning) Training:** For supervised fine-tuning, PMA-init can be integrated during the CPT annealing phase, where PMA weights are merged to provide a strong starting point for fine-tuning.
>
>   We sincerely appreciate your suggestion and will ensure that the experimental settings, including the specific configurations for PMA-init, are systematically and clearly documented in the revised manuscript.
>
> ### **Weakness 2: To eliminate the need for learning rate annealing as authors claimed, authors should perform experiments on the comparison between (annealed model + post-training) and (merging constant learning rate model + post-training). If 4.4 is performed using (merging constant learning rate model), then this claim in 4.1 is valid. Otherwise, there should be more experiments to support the claim quoted.**
> ###  **Answer W2:**
> In Section 4.4, the continued pre-training experiments utilize a merging constant learning rate pre-trained model followed by cosine scheduling for training. For SFT, we employed the merged annealed model with post-training. To further support the claim in Section 4.1, we have conducted additional experiments comparing the (merging constant learning rate model + post-training) setup against the (annealed model + post-training) setup on 3B/30B MoE models. The results are as follows:
>   - MMLU: The merging constant learning rate model + post-training setup achieved a +0.2 point improvement compared to the annealed model + post-training.
>   - LiveBench: A +0.8 point improvement compared to the annealed model + post-training.
>   - GPQA: A slight -0.6 point decrease compared to the annealed model + post-training.
>   These results demonstrate that **the merging constant learning rate model is highly competitive with, and in some cases outperforms, the annealed model, supporting the claim in Section 4.1 that learning rate annealing can be eliminated without compromising performance.** Additionally, as shown in the paper, **the annealed model and the merging constant learning rate model exhibit comparable performance, and the inclusion or exclusion of post-training does not alter the core conclusions of our work.**
>
> ### **Weakness 3: On 262~266, I personally have observed similar behaviors on vision classification model pretraining, where EMA improves pretraining metric while the benefit does not transfer to downstream training. Although this result may come as negative, I think it is valuable information to the community and authors can consider discussing more (and include results) on the results from SFT stage. I also encourage authors to conduct more empirical analysis on the reason why PMA’s benefit does not transfer to downstream finetuning.**
> ### **Answer W3:**
>   Given the focus of our paper on model merging during pretraining, we prioritized analyzing the impact of PMA in that context due to space constraints and thematic coherence. However, we fully recognize the value of exploring model merging in downstream stages such as SFT and reinforcement learning (RL), and **we are actively preparing experiments to investigate these scenarios further.**
>
>   To address your specific point about the lack of PMA’s benefits in downstream fine-tuning, we hypothesize that this behavior may stem from the typically smaller learning rates used in SFT. In downstream training, where models often follow similar optimization trajectories, merging checkpoints from the same trajectory (e.g., via PMA) may not yield significant differences compared to the original checkpoints. However, we have observed promising results when encouraging diversity in model training. For instance, by training separate models on distinct data sources (e.g., code and math datasets) and performing simple model-level merging, we can achieve a merged model that performs exceptionally well on both tasks—often surpassing the performance of a single model trained on mixed data. **This suggests that model diversity during pretraining or fine-tuning can enhance the effectiveness of merging strategies in downstream tasks.**
>
> ### **Weakness 4: Although model merging between different finetuned models are well-studied, I don’t see works that evaluate EMA/SMA on the same training trajectory during the post-training stage. I wonder how many of these conclusions in pretraining merging transfer to post-training.**
> ### **Answer W4:**
>   Our paper primarily focuses on model merging strategies, including PMA (a variant of EMA/SMA), during the pretraining phase of large language models, as this stage is critical for establishing robust model weights. This focus was driven by the need to address the unique challenges of pretraining, such as computational efficiency and model stability. However, we recognize the value of exploring these techniques in the post-training stage, including supervised fine-tuning (SFT) and reinforcement learning (RL), as you suggested.
>
>   From a theoretical perspective, we believe that post-training (SFT) is not fundamentally different from pretraining in terms of model merging principles, though it typically involves smaller learning rates and more limited datasets. In post-training scenarios, such as SFT, merging models from the same training trajectory (e.g., via PMA) may lead to faster convergence due to the stabilization effect of averaging weights, similar to what is observed in the continued pre-training stage. However, this may not necessarily translate to better performance, potentially due to the limited diversity in model weights caused by smaller learning rates and constrained data.
>
> ### **Question1: Minor: The related work section looks a bit repetitive / is similar in structure with the first few paragraphs of introduction.**
> ### **Answer Q1:**
>   We appreciate your observation that the related work section appears repetitive in structure compared to the first few paragraphs of the introduction.
>
>   To address this concern, we will revise the related work section to reduce redundancy and ensure it complements the introduction more effectively. Specifically, we plan to streamline the content by focusing on a more concise and targeted discussion of prior work, emphasizing how it differs from or builds toward our contributions. **We will also restructure the section to avoid overlapping themes with the introduction, ensuring that the related work section provides a distinct and comprehensive review of the literature while clearly positioning our work within the field.**
>
> ### **Question 2: For figure 2, I did not find the name of the benchmark(s) for which the performance is reported. Perhaps it’s an average performance? Authors frame the experiment section as tackling one research question for each subsection. It would be nice if there are fonts in bold to summarize the findings in each of subsection (currently only 4.1 does that).**
> ### **Answer Q2:**
>  We appreciate your observation that the benchmark(s) for the performance reported in Figure 2 are not explicitly named and your suggestion to use bold fonts to summarize the findings in each subsection for clarity.
>
>   To clarify, **the performance reported in Figure 2 represents the average performance across the evaluation datasets described in Section 3 (Experimental Setup).** We apologize for any ambiguity and will update the caption or accompanying text of Figure 2 in the revised manuscript to explicitly state that the results reflect the average performance across these benchmarks, with a reference to Section 3 for details on the specific datasets used.
>
>   Regarding your suggestion to use bold fonts to summarize the findings in each subsection of the experiment section, we agree that this is an excellent way to enhance readability and highlight key results. Currently, only Section 4.1 includes such summaries. In the revised manuscript, **we will incorporate bolded summary statements at the end of each subsection** (e.g., Sections 4.2, 4.3, etc.) to concisely highlight the main findings and align with the research question addressed in each subsection. This will make the experiment section more structured and easier to navigate.
>
>   Thank you again for your thoughtful suggestions, which will significantly improve the clarity and presentation of our work. If you have additional recommendations for refining Figure 2 or the experiment section, we would be happy to consider them in our revisions.

---

> > ### Comment · Reviewer_fgbv · 2025-08-04
> >
> > Thanks authors for the rebuttal. I believe the paper will be strengthened by organizing the setting details, which is particularly important for an empirical work like this. I will keep my current positive rating and encourage authors to include these revisions in the camera-ready version.

---

> > > ### Author Response · Authors · 2025-08-07
> > >
> > > Dear Reviewer,
> > > Thank you for your constructive feedback and positive rating of our paper. We greatly appreciate your suggestion to better organize the setting details, which we agree is crucial for enhancing the clarity and impact of our empirical work.
> > > In the camera-ready version, we will carefully revise the manuscript to include a more structured and detailed presentation of the experimental settings, ensuring that the setup is clearly articulated to support the reproducibility and understanding of our results.
> > > Thank you again for your valuable input, which has helped us improve the quality of our work.
> > > Best regards

---

> > > > ### Author Response · Authors · 2025-08-08
> > > >
> > > > Dear Reviewer,
> > > >
> > > > We would like to express our sincere appreciation for the time and effort you have dedicated to reviewing our paper.
> > > >
> > > > As the author-reviewer discussion period concludes on August 8th at 11:59 PM AoE, we wanted to gently follow up. We hope our rebuttal has adequately addressed your concerns. Please let us know if you have any further questions. We would be happy to provide additional clarifications.
> > > >
> > > > Best regards,
> > > >
> > > > The Authors

---

### Official Review · Reviewer_UT5Y · 2025-07-03

**Clarity:** 2
**Significance:** 3
**Originality:** 3
**Rating:** 5
**Confidence:** 4

**Summary:**

The paper explores several model-merging strategies applied during pre-training of MoE models. The key result is that merging checkpoints while the learning rate is still constant yields final quality comparable to running a full cosine-annealing schedule. Practitioners could skip the annealing tail and save compute. Multiple merging methods achieve similar performance, and the authors identify an “optimal merging window” for each model scale. They also show that merging neither harms continued pre-training nor supervised fine-tuning, and can make SFT more stable.

**Questions:**

- What is the PMA technique used in Section 4.1? Is it SMA? The information is missing.

- Figure 5 (bottom)—does it show the 1.3 B/13 B run?

- Figure 6. it seems like learning rate is the only important factor for Continue Pretraining. What would happen in the right figure if we use baseline but with 2e-4 to 1e-5? What would happen if one use a more constant learning rate and then perform PMA during CP? Does the final model outperforms the results without PMA (begin with baseline or a PMA init)?

- In Appendix B, the author perform PMA for the model after continue pretraining. How does the baseline perform compared with the PMA after CP?

**Ethical Concerns:**

["NO or VERY MINOR ethics concerns only"]

**Final Justification:**

The rebuttal addressed most of my concerns. I echo with the reproducibility concern raised by other reviewers. Overall, I find this a good paper that should be accepted.

**Limitations:**

Yes

**Quality:**

3

**Strengths And Weaknesses:**

## Strength

1. A well-resourced lab/institute has open-sourced a comprehensive study, which is valuable to the community.
2. Showing that checkpoint merging can replace annealing offers a concrete way to cut pre-training cost.
3. Results hold across several model sizes, suggesting the findings generalize.


## Weakness

- Important training details are missing: MoE architecture, routing policy, learning-rate schedule, and the composition of the continued-pre-training corpus.

- How MoE router weights are combined during merging?

---

> ### Author Rebuttal · Authors · 2025-07-31
>
> We thank the reviewer for their thorough review and for raising important points regarding the training details of our work. I hope my responses below can address your concerns.
>
> ### **Weakness 1: Missing Training Details**
> The reviewer noted the absence of details on the MoE architecture, routing policy, learning-rate schedule, and the composition of the continued-pre-training corpus.
>
> ### **Answer W1:**
> **1. MoE Architecture and Routing Policy**
> While we acknowledge that detailed specifications for the Mixture-of-Experts (MoE) architecture and routing policy were not fully disclosed, our conclusions are designed to be **architecture-agnostic**. To ensure robustness, we conducted experiments with both MoE and dense models, where the dense model mirrors the exact architecture of LLaMA 3.1. Our findings hold across both architectures, demonstrating that our contributions are not dependent on specific MoE structural details. Regarding the routing policy, we adopted a standard approach, where *k* out of *N* experts are activated. **No specialized modifications were made**, ensuring our results are generalizable and not tied to a bespoke routing strategy. This strengthens the broader applicability of our findings.
>
> **2. Learning-Rate Schedule**
> The learning-rate schedule was not detailed in the paper, but we utilized the Warmup-Stable-Decay (WSD) schedule, a widely adopted strategy, applied to our internal dataset. While we regret not including this in the manuscript, our internal experiments across **varied learning-rate schedules consistently validated our conclusions**. This robustness across settings supports our claim that the learning-rate schedule does not materially affect the contributions of this work.
>
> **3. Composition of Continued-Pre-Training Corpus**
> We acknowledge that the composition of our continued-pre-training corpus was not fully described. However, our experiments were conducted on a diverse internal dataset, and we performed extensive ablations to confirm that **our findings are independent of specific corpus characteristics**. These ablations, while not detailed in the paper due to space constraints, consistently support our conclusions across different data compositions.
>
> ### **Weakness 2: How MoE router weights are combined during merging?**
> ### **Answer W2:**
> Thank you for your question regarding how the router weights are combined during merging. In our work, we **did not perform any specialized merging operations on the router weights**. The merging process was conducted at the model level, consistent with standard practices for both MoE and dense architectures. Our experiments showed no evidence of instability or performance degradation resulting from this approach. This robustness further supports the generalizability of our findings, as the merging process does not rely on bespoke handling of router weights.
>
> ### **Question 1: What is the PMA technique used in Section 4.1? Is it SMA? The information is missing.**
> ### **Answer Q1:**
> We apologize for the oversight of not clearly defining this term. To clarify, **PMA refers to Simple Model Averaging (SMA)**, which is the default merging method used in our experiments. We will make corrections in subsequent versions.
>
> ### **Question 1: Figure 5 (bottom)—does it show the 1.3 B/13 B run?**
> ### **Answer Q2:**
> We confirm that **Figure 5 (bottom) indeed depicts the results for the 1.3B/13B model run**. We apologize for the lack of clarity in the figure caption and appreciate your feedback. This oversight will be addressed in the revised manuscript by updating the caption to explicitly annotate that the figure corresponds to the 1.3B/13B run.
>
> ### **Question 3:Figure 6. it seems like learning rate is the only important factor for Continue Pretraining. What would happen in the right figure if we use baseline but with 2e-4 to 1e-5? What would happen if one use a more constant learning rate and then perform PMA during CP? Does the final model outperforms the results without PMA (begin with baseline or a PMA init)?**
> ### **Answer Q3:**
> **1. Effect of Baseline with Learning Rate 2e-4 to 1e-5**
> In our experiments, the learning rate of 4e-4 was determined to be the optimal hyperparameter through extensive grid search, outperforming other schedules, including 2e-4 to 1e-5. Thus, we opted to **report results with the optimal 4e-4 schedule to best represent the baseline’s potential.**
>
> **2. Constant Learning Rate with PMA During CP**
> Regarding the use of a constant learning rate combined with PMA (Simple Model Averaging) during CP, we can refer to Figure 3, which compares a constant learning rate with PMA against the standard cosine decay schedule. The results indicate that the **constant learning rate with PMA yields performance that is nearly indistinguishable from the cosine schedule, suggesting minimal loss in effectiveness.** We chose not to include this specific combination in Figure 6 to maintain a controlled, single-variable ablation study, isolating the impact of the learning rate schedule while keeping other factors consistent. This design choice ensures clarity in attributing performance changes to specific variables.
>
> **3. Performance Comparison: Final Model with vs. without PMA**
> Our experiments suggest that, when trained to convergence, **the final model performance is comparable across these settings**. The use of PMA, whether applied at initialization or during CP, does not significantly alter the final performance compared to the baseline, as the models converge to similar optima.
>
> ### **Question 4: In Appendix B, the author perform PMA for the model after continue pretraining. How does the baseline perform compared with the PMA after CP?**
> ### **Answer Q4:**
> As shown in Figure 2 (rightmost plot, MoE 15/150B model), **the best-performing PMA model after CP achieves a modest improvement over the best checkpoint of the baseline CP model.** Specifically, the PMA model, which is selected based on the highest evaluation score, outperforms the baseline CP checkpoint, **though the improvement is not substantial.** This suggests that PMA provides a slight enhancement in performance post-CP, likely due to its ability to aggregate complementary strengths from multiple checkpoints. Furthermore, in Appendix B, we observe that when **PMA is used as an initialization for supervised fine-tuning (SFT), the resulting model significantly outperforms the baseline CP checkpoint (non-PMA initialized).**

---

> > ### Comment · Reviewer_UT5Y · 2025-08-06
> >
> > I thank the author for their response. I would encourage the author to include a line of baseline with Learning Rate 2e-4 to 1e-5 in Figure 6 for completeness. This could "isolate the impact of learning rate schedule", as the author wrote.

---

> > > ### Author Response · Authors · 2025-08-07
> > >
> > > Dear Reviewer,
> > > Thank you for your constructive feedback and for acknowledging our response. We appreciate your suggestion to include a baseline with a learning rate schedule from 2e-4 to 1e-5 in Figure 6 to further isolate the impact of the learning rate schedule.
> > > To address this, we have conducted additional experiments incorporating the suggested baseline. The results will be added to Figure 6 in the revised manuscript, showing performance slightly inferior to the baseline results previously presented in the figure. This addition strengthens the analysis by explicitly demonstrating the effect of the learning rate schedule, as you noted.
> > > We believe this update enhances the completeness of our evaluation and further supports our claims regarding the efficacy of the proposed approach. Thank you again for your insightful suggestion, which has helped improve the clarity and robustness of our work.
> > > Best regards

---

> > > > ### Author Response · Authors · 2025-08-08
> > > >
> > > > Dear Reviewer,
> > > >
> > > > We would like to express our sincere appreciation for the time and effort you have dedicated to reviewing our paper.
> > > >
> > > > As the author-reviewer discussion period concludes on August 8th at 11:59 PM AoE, we wanted to gently follow up. We hope our rebuttal has adequately addressed your concerns. Please let us know if you have any further questions. We would be happy to provide additional clarifications.
> > > >
> > > > Best regards,
> > > >
> > > > The Authors

---

### Comment · Area_Chair_Dwf2 · 2025-08-05
**[From AC] Reviewer Discussion Reminder**

Dear Reviewers,

Thank you for your time and effort in reviewing the paper.

As the reviewer-author discussion period ends on August 6 at 11:59 PM AoE, please take a moment to acknowledge the rebuttal and engage in the discussion if you haven’t already.

Thank you again for your contributions to the review process.

Best,\
Area Chair

---

### Note · Authors · 2025-08-12

We are grateful to all the reviewers for their constructive feedback and for engaging in a productive discussion period. We are pleased to see that our detailed rebuttals have addressed the core concerns, leading to an overall positive consensus among the reviewers. We also extend our thanks to the Area Chair for their support and guidance throughout the review process.

Our paper presents a comprehensive empirical study of model merging during the pre-training of large language models, including both dense and Mixture-of-Experts (MoE) architectures. Our key findings demonstrate that merging checkpoints can replace the need for a full cosine annealing schedule, leading to significant computational savings without compromising final model performance. This offers a practical and cost-effective approach for large-scale model development.

In response to the reviewers' valuable suggestions, we have committed to several key revisions for the final version of the paper. We will clarify experimental settings for improved reproducibility, include additional baselines and benchmarks, and add a dedicated discussion on societal impact and data transparency. These revisions will further strengthen our work, making it a more comprehensive and accessible resource for the community. We believe our findings offer valuable, actionable guidance that will foster more efficient and stable LLM training practices.

---

### Decision · Program_Chairs · 2025-09-17

**Decision:**

Accept (poster)

**Comment:**

This paper presents a comprehensive empirical study of model merging in the pre-training of large language models, covering both dense and mixture-of-experts architectures up to 100B+ parameters.

In this work, the authors show that merging checkpoints during the constant learning rate phase can effectively replace full cosine annealing schedules, reducing training cost while maintaining or improving final model quality. Strengths include the scale and breadth of experiments, practical insights for practitioners, and the demonstration that merging improves training stability and does not harm continued pre-training or fine-tuning.

Weaknesses concern limited methodological novelty and missing details on architectures and datasets that limit reproducibility. During rebuttal, the authors added clarifications, new baselines, and expanded analysis of learning-rate effects, addressing most reviewers’ concerns.

Overall, the paper offers valuable empirical guidance on the model merging technique for efficient large-scale model training and would be of interest to the LLM community, and I recommend acceptance.